


# Tropospheric transport and unresolved convection: numerical experiments with CLaMS-2.0/MESSy

Paul Konopka[1], Mengchu Tao[2], Marc von Hobe[1], Lars Hoffmann[3], Corinna Kloss[1*],
Fabrizio Ravegnani[4], C. Michael Volk[5], Valentin Lauther[5], Andreas Zahn[6], Peter Hoor[7], and
Felix Ploeger[1]

[1]Forschungszentrum Jülich, IEK-7, Germany
[2]Carbon Neutrality Research Center, Institute of Atmospheric Physics, Beijing, China
[3]Jülich Supercomputing Centre, Forschungszentrum Jülich, Germany
[4]National Research Council - Institute for Atmospheric Sciences and Climate (ISAC-CNR), 40129 Bologna, Italy
[5]Institute for Atmospheric and Environmental Research, University of Wuppertal, Wuppertal, Germany
[6]The Institute of Meteorology and Climate Research (IMK), Karlsruhe Institute of Technology, Karlsruhe, Germany
[7]Institute for Atmospheric Physics, Johannes Gutenberg University, Mainz, Germany
[*]now at: Laboratoire de Physique et Chimie de l'Environnement et de l'Espace (LPC2E), Universitè d'Orlèans, France

*Correspondence to:* Paul Konopka (p.konopka@fz-juelich.de)

**Abstract.** Pure Lagrangian, i.e. trajectory-based transport models, take into account only the resolved advective part of transport. That means neither mixing processes between the air parcels (APs) nor unresolved subgrid-scale advective processes like convection are included. The Chemical Lagrangian Model of the Stratosphere (CLaMS-1.0) extends this approach by including mixing between the Lagrangian APs parameterizing the small-scale isentropic mixing. To improve model representation

of the upper troposphere and lower stratosphere (UTLS), this approach was extended by taking into account parameterization of tropospheric mixing and unresolved convection in the recently published CLaMS-2.0 version. All three transport modes, i.e. isentropic and tropospheric mixing as well as the unresolved convection can be adjusted and optimized within the model. Here, we investigate the sensitivity of the model representation of tracers in the UTLS with respect to these three modes.

For this reason, the CLaMS-2.0 version implemented within the Modular Earth Submodel System (MESSy), CLaMS-

2.0/MESSy, is applied with meteorology based on the ERA-Interim (EI) and ERA5 (E5) reanalyses with the same horizontal resolution ($1.0 \times 1.0$ degree) but with 60 and 137 model levels for EI and E5, respectively. Comparisons with in situ observations are used to rate the degree of agreement between different model configurations and observations. Starting from pure advective runs as a reference and in agreement with CLaMS-1.0, we show that among the three processes considered, isentropic mixing dominates transport in the UTLS. Both the observed CO, $O_3$, $N_2O$ and $CO_2$ profiles as well as CO-$O_3$ correlations are clearly

better reproduced in the model with isentropic mixing. The second most important transport process considered is unresolved convection. This additional pathway of transport from the Planetary Boundary Layer (PBL) to the main convective outflow dominates the composition of air in the lower stratosphere relative to the contribution of the resolved transport. This transport happens mainly in the tropics and sub-tropics, and significantly rejuvenates the age of air in this region. By taking into account tropospheric mixing, weakest changes in tracer distributions without any clear improvements were found.





# 1 Introduction

Timescales of transport from the Earth's surface into the upper troposphere and lower stratosphere (UTLS) determine chemical composition of this region and, consequently, influence the Earth's radiation budget and surface temperatures (Riese et al., 2012). The Lagrangian, i.e., trajectory-based, view of transport has proven to successfully resolve both small-scale structures
as well as signatures of long-range transport, mainly due to a reduced numerical diffusion compared to the Eulerian-based transport models (see e.g. Lin et al. (2013) and references therein). However, trajectory-based (Lagrangian) models, which are driven by resolved advective winds as provided by meteorological reanalyses, still miss important physical drivers of transport like unresolved (subgrid-scale) convection and all mixing processes including isentropic or other, small-scale, mainly tropospheric mixing processes. While parameterization of unresolved convection can be understood as an extension of the advective
part of transport (additional up- and downdrafts), including of mixing requires an extra mass transfer between the air parcels (APs), the latter being always a challenge in the irregular grid of Lagrangian APs. In general, motion in the free atmosphere can be divided into two parts: the 2d isentropic transport on surfaces with constant, dry or moist, potential temperature and the cross-isentropic transport, roughly perpendicular to such surfaces. Sufficiently far away from the Planetary Boundary Layer (PBL) and from regions affected by strong convection, the cross-isentropic transport, mainly driven by radiation, is much slower
compared to fast isentropic transport typically disturbed by all types of waves and finally leading to irreversible, isentropic mixing. In addition, both PBL and convection are drivers of tropospheric mixing and have to be appropriately parameterized not only in the Lagrangian transport models.

Currently, there are only few Lagrangian transport models that include a parametrization of unresolved convection and mixing (Brinkop and Jöckel, 2019; Wohltmann et al., 2019; Konopka et al., 2019). Here we use the recently published CLaMS-
20 2.0 version (Konopka et al., 2019), which extends the concept of isentropic mixing (CLaMS-1.0) by including tropospheric mixing and a simple parameterization of convective uplifts. In this paper, we aim to quantify, the relative importance of these three unresolved transport modes using as a reference pure, trajectory-based studies. Extending the approach presented in Konopka et al. (2019) and Wohltmann et al. (2019), we do it on a global scale and over time periods of more than ten years in order to find the impact of these new, rather tropospheric transport modes, even on the stratospheric circulation. For
validation, idealized tracers as well as comparisons with in-situ observations of CO, $O_3$, $N_2O$ and $CO_2$ are investigated. Using reanalysis-driven (ERA-Interim and ERA5) CLaMS studies we also aim to enhance the overall agreement of the model with the observations extending in this way the Brinkop and Jöckel (2019) results where Lagrangian transport within a climate model framework was investigated. Our main focus is to quantify the role of unresolved convective updrafts and of tropospheric mixing on the representation of climate-relevant trace gases in the UTLS region.

# 2 Transport sensitivity scenarios

The setup of CLaMS-2.0/MESSy follows the CLaMS-2.0 configuration described in Konopka et al. (2019) and uses the Modular Earth Submodel System (MESSy) as a software framework (Jöckel et al., 2010) to run the model on the supercomputer JUWELS (Jülich Supercomputing Centre, 2019). To simplify the notation, we omit the suffix MESSy in the following. ERA-





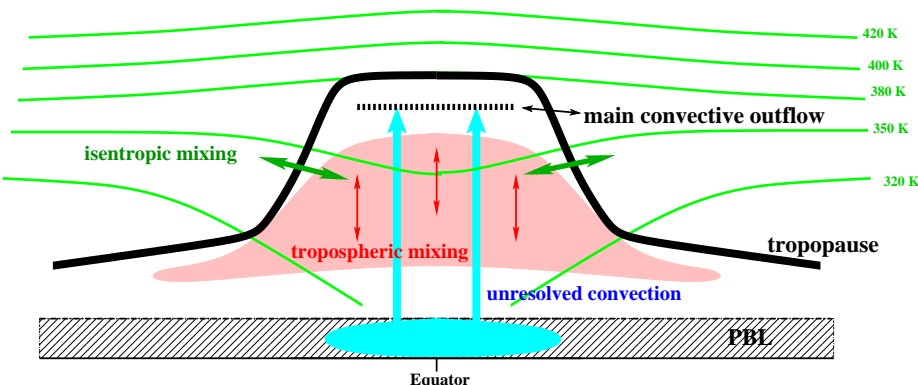

**Figure 1.** In CLaMS-2.0, an interplay between three types of extended transport is investigated: isentropic mixing (CLaMS-1.0), tropospheric mixing and unresolved convective updrafts (CLaMS-2.0). Extended transport means an extension relative to a pure trajectory-based transport that is used in this study as a reference.

Interim (EI) and ERA5 (E5) reanalyses from the European Centre for Medium-Range Weather Forecasts ECMWF) provide the meteorology to drive the model (Dee et al., 2011; Hersbach et al., 2020). Although the same horizontal resolution ($1.0 \times 1.0$ degree) is used for both data sets, the vertical resolution is different, with 60 and 137 levels for EI and E5, respectively, corresponding to the maximal vertical resolution available for these products. The vertical layer depths at 5 to 20 km of altitude

vary between 0.3 to 0.4 km for E5 and 0.5 to 1.4 km for EI (Hoffmann et al., 2019). Hoffmann et al. (2019) also found that transition from EI to E5 improves the Lagrangian transport simulations mainly due to improved representation of convective updrafts.

   The baseline for the following comparisons is a CLaMS reference simulation (CLaMS-1.0, set-up described by Pommrich et al. (2014)), which covers the period 1979-2017 and is avaiable in two versions driven by EI and E5, respectively. All CLaMS-2.0

simulations start on 01.01.2017, use CLaMS-1.0 for initialization and cover one year. We choose 2017 because data of two major aircraft campaigns covering tropics and extra-tropics are avaiable (StratoClim and WISE, see section 3.2) as well as the reanalyses products EI and E5 can be downloaded from the ECMWF server (EI only until mid of 2019). To check the long-term effects on tracer distributions for different model configurations described below, perpetuum runs are performed (14-times 2017) as described in Konopka et al. (2017) and Poshyvailo et al. (2018). Perpetuum runs approximate transient

runs, which are numerically more expensive. A short description of the differences between CLaMS-1.0, CLaMS-2.0 and the slightly modified CLaMS-2.0 version used here can be found in Table 1. Details of these modifications are explained in the Appendix.

   By redefining the parameters listed in Table 1, CLaMS-2.0 can be run as CLaMS-1.0. Thus, in CLaMS-2.0, there is an interplay between three parameterized components of transport: isentropic mixing (I), unresolved convection (C) and tropospheric mixing (T) schematically shown in Fig. 1. In the following, configurations with respective components switched off and on

are denoted with "0" and "1", respectively. Because we do not carry out any systematic sensitivity of their strength, the choice





| | CLaMS-1.0 (Pommrich et al., 2014) | CLaMS-2.0 (Konopka et al., 2019) | CLaMS-2.0 (this paper) |
|---|---|---|---|
| lowest layer (PBL) | $\Delta\zeta = 100$K<br>$r = 110$km | $\Delta\zeta = 140$K<br>$r = 110$km | $\Delta\zeta = 140$K<br>$r = 90$km |
| boundary conditions | every 24 h | every 6 h | every 6 h |
| isentropic mixing | every 24 h<br>$\lambda = 1.5\,\mathrm{d}^{-1}$<br>$\epsilon = 1.0$ | every 6 h<br>$\lambda = 3.5\,\mathrm{d}^{-1}$<br>$\epsilon = 1.0$ | every 6 h<br>$\lambda = 4\,\mathrm{d}^{-1}$<br>$\epsilon = 0.3$ |
| vertical mixing | no | $N_d^2 = 0$ | variable $N_d^2$ |
| unresolved convection | no | $N_m^2 = 0$<br>$\Delta\theta_c = 30$ K | variable $N_m^2$<br>variable $\Delta\theta_c$ |

**Table 1.** Changes between CLaMS-1.0 (1st column), CLaMS-2.0 (2nd column) and CLaMS-2.0 version used in this paper (3rd column, for details see Appendix). Notation: $\Delta\zeta$ (in K) - thickness of the lowest model layer approximating the Planetary Boundary layer (PBL); values of $\Delta\zeta = 100, 140$ K correspond to a maximal geometric thickness of 2 and 3 km, respectively, of this orography following layer. $r$ (in km) - horizontal mean distance between the APs in the lowest model layer, $\lambda$ (in $\mathrm{d}^{-1}$) - (critical) Lyapunov exponent, $\epsilon$ - merging parameter, $N_d^2$, $N_m^2$ (in $\mathrm{s}^{-2}$) - (critical) dry and moist Brunt-Vaisala frequency, $\Delta\theta_c$ (in K) (critical) vertical uplift of convective updrafts. $\theta, \zeta$ - potential and hybrid potential temperature (Mahowald et al., 2002). In CLaMS-2.0 version used here, the parameters of isentropic mixing were refined to reproduce better the intensity of isentropic mixing used in CLaMS-1.0. Furthermore, the horizontal resolution in the lowest layer was sligtly enhanced (from 110 to 90 km) to fit better the input reanalysis.

"1" means a configuration with rather too strong contribution of the respective component. For comparison, CLaMS-1.0 runs are used (C-1.0). In addition, CLaMS-1.0 runs including the same definition of the lower boundary as in CLaMS-2.0 will be abbreviated with "C-1.2". All performed runs are listed in Table 2. If all transport extensions are switched on (i.e. for I1, C1, T1), the following values of the model parameters introduced in Table 1 are used:

I1    :    $\lambda = 4.0\,\mathrm{d}^{-1}$,    mixing frequency = 6 hours    $\epsilon = 0.3$

   C1    :    $N_m^2 = 0.0 \times 10^{-4}\,\mathrm{s}^{-2}$,    $\Delta\theta_c = 10$ K

   T1    :    $N_d^2 = 1.0 \times 10^{-4}\,\mathrm{s}^{-2}$

Thus, isentropic mixing (I) is controled by the Lyapunov exponent $\lambda$ and the mixing frequency $\Delta t$ defining the critical deformations in the flow, $\lambda\Delta t$, as well as the merging parameter $\epsilon$ controlling the intensity of the grid adaption procedure including
new air APs into the flow (see Appendix for more details). The different choice of mixing paramaters compared to CLaMS-1.0 is owed to the fact that in order resolve the diurnal cycle of convective updrafts, higher mixing frequency, without any significant change of the total intensity of mixing, is required. (every 6 hours in CLaMS-2.0 instead of every 24 hours in CLaMS-1.0). Thus, the convection parametrization (C) is called every 6 hours in CLaMS-2.0. Its intensity is regulated by the moist Brunt-Vaisala frequency $N_m^2$ triggering the onset of convection in the the PBL and the minimal vertical uplift $\Delta\theta_c$ deciding which convective updrafts are taken into account. Finally, the dry Brunt-Vaisala frequency $N_d^2$ triggers the onset of tropospheric mix-



| Type | Short Name | Remarks |
|---|---|---|
| climat. runs 1979-2017 | C-1.0-EI | CLaMS-1.0/EI |
| | C-1.0-E5 | CLaMS-1.0/E5 |
| | C-1.2-EI | like C-1.0-EI but lowest layer (PBL) like in CLaMS-2.0 |
| CLaMS-2.0 2017 | I1C1T0-EI | best choice |
| | I1C1T0-E5 | |
| sensitivity runs, EI | I1C0T0-EI | isentr. mixing |
| | I1C0T1-EI | isentr. + trop. mixing |
| | I1C1T0-EI | isentr. mixing + unresol. conv. |
| | I1C1T1-EI | isentr. mixing + unresol. conv. + trop. mixing |
| pure advection | I0C0T0-EI | 54 instead of 45 layers to get similar # of APs |
| sensitivity runs, E5 | I1C1T0-E5 | best choice (like above) |
| | I1C1(30K)T0-E5 | less convection ($\Delta\theta_c = 30$ instead 10 K) |
| | I1C1(-1.0)T0-E5 | less convection ($N_m^2 = -1.0 \times 10^{-4}$ instead 0 s$^{-2}$) |
| | I1C1(-1.0/30K)T0-E5 | much less convection ($\Delta\theta_c = 30$ K and $N_m^2 = -1.0 \times 10^{-4}$ s$^{-2}$) |
| | I1C0T0-E5 | only isentropic mixing (no unresolved convection, reference) |

**Table 2.** List of performed runs. For used notation see text. All runs are performed within the software framework MESSy and are also avaiable als perpetuum runs (14-times 2017).

ing by averaging the mixing ratios of a considered AP with all its next neigbours. A stably stratified stratosphere, with typical values of the Brunt-Vaisala frequency $N^2$ much larger than $N_d^2$, prevents this type of mixing. We coined this type of mixing "tropospheric mixing" because it happens in the model only below the tropopause (Konopka et al., 2019).

## 3 Results

To measure the quality of transport, an e90-tracer-based diagnostic will be applied in a first step (Prather et al., 2011; Abalos et al., 2017). In addition, we compare the simulated distributions of CO, $O_3$, $N_2O$, $CO_2$ with in situ observations to evaluate different transport scenarios. In particular, metrics based on the differences to the observed time series as well as on differences between the observed and simulated CO-$O_3$ correlations will be used (Konopka et al., 2004; Wohltmann and Rex, 2009) to rate an overall agreement between the observations and the model. All the available in-situ data are used for this without any presorting, filtering or weighting.

### 3.1 e90 tracer

The e90 tracer is an idealized tracer, uniformly distributed at the Earth's surface, with a lifetime of 90 days which is long relative to the time scale of vertical transport in the troposphere but short compared to the time scale of vertical transport in

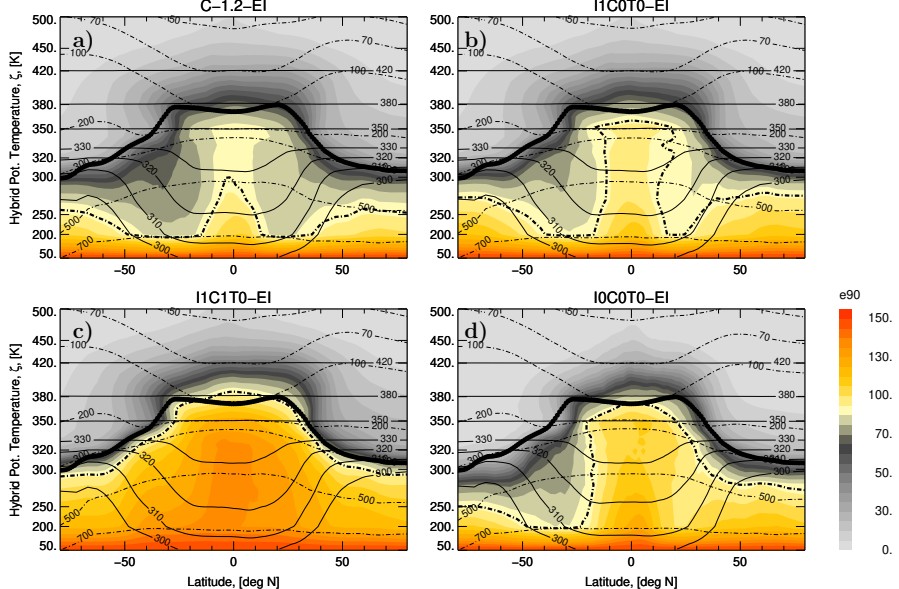

**Figure 2.** Zonal and monthly mean (December 2017) of the e90-tracer distribution for 4 CLaMS/EI configurations: with only isentropic mixing in CLaMS-1.2 (a) and 2.0 (b), with isentropic mixing and unresolved convective updrafts in ClaMS-2.0 (c) and pure advection calculation in ClaMS-2.0 (d). The black dashed and solid lines are the 90 ppbv isoline of the e90 tracer and the WMO tropopause, respectively.

the stratosphere. Because e90 tracer mimics well the spatial and temporal distribution of carbon monoxide (CO) (Prather et al., 2011), we set this tracer to 150 ppbv in the lowest layer of CLaMS approximating the well-mixed PBL. Thus, following Prather et al. (2011), large vertical gradients across the WMO tropopause were found in the model with the 90 ppbv isoline well approximating both the chemical (100 ppb $O_3$) and the WMO tropopause.

In Fig. 2, four CLaMS/EI configurations are shown (zonal and monthly mean for December 2017). Isentropic mixing both in CLaMS-1.2 (Fig. 2a) and 2.0 (Fig. 2b) increases downward transport of stratospheric air into the troposphere if compared with pure advective transport (Fig. 2d). In the full extension of CLaMS-2.0 (Fig. 2c), there is the smallest distance between the e90 isoline and the WMO tropopause. Note that the same definition of the lowest layer approximating the PBL was used in the CLaMS-1.2 and 2.0 simulation (Fig. 2a, b). The only difference between these two runs is the update frequency of the boundary conditions (24 versus 6 hours in the CLaMS-1.2 and 2.0, respectively). Thus, in CLaMS-2.0, the lowest boundary is reset more frequently (i.e. all APs in this layer are set to their default positions every 6 hours) and, mainly in the tropics, the resolved convective uplifts can be better sampled increasing upward transport and shifting upwards the 90 ppbv isoline (Fig. 2b versus 2a).

An interesting feature can be seen when isentropic mixing is added to a pure advective transport (from Fig. 2d to 2b). The boreal winter isentropic mixing in the NH seems to push down the 90 ppbv isoline, mainly in the extratropics where a



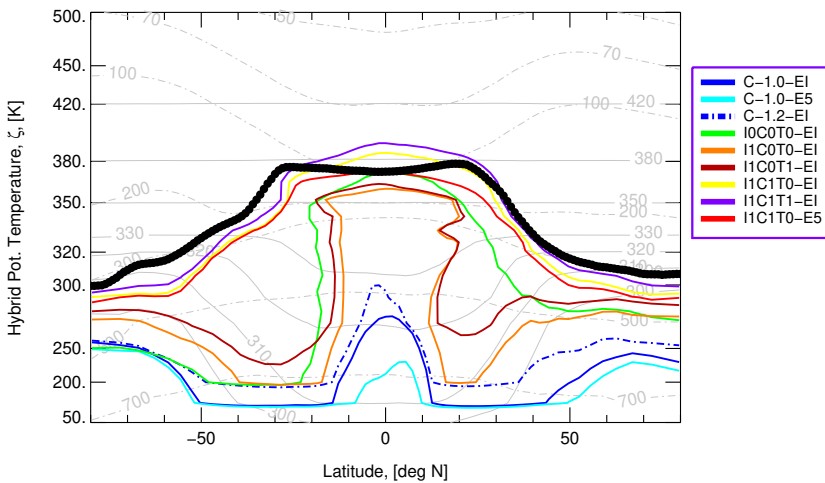

**Figure 3.** Like in Fig. 2 but only the 90 ppbv isoline of the e90 tracer is color-coded for runs listed in Table 2. Thick solid black line denotes the WMO tropopause.

strong bi-directional isentropic transport between the lowermost stratosphere and upper tropical troposphere is expected. Due to aging by (isentropic) mixing (Garny et al., 2014), the re-circulated air within the Hadley cell and within the lower branch of the Brewer-Dobson circulation in the NH becomes older, shifting downwards the 90 ppbv isoline. In the SH, the reverse

5   process during the Austral summer moves the 90 ppbv isoline upwards due to mixing younger tropical air into the polar region. Finally, unresolved convection seems to work against the effect of aging by mixing and lifts the 90 ppbv isoline upwards into the region around the tropopause (Fig. 2c).

A more systematic comparison of all runs is shown in Fig. 3. The lowest 90 ppbv isoline can be seen for the C-1.0-E5 setup while the highest 90 ppbv isoline can be diagnosed for the EI-run with all unresolved transport parameterizations switched on

10   (i.e. I=1, C=1, T=1). The largest deviation of the 90 ppbv isoline from the WMO tropopause can be diagnosed for the baseline runs with a small improvement by moving from the C-1.0 to C-1.2 configuration. The best e90-performance is obtained for the I1C1 runs. If tropospheric mixing is additionally switched on (I1C1T1), the 90 ppbv isoline moves above the tropical tropopause suggesting too much upward propagation of troposheric signatures in such a model configuration. Comparison with the I0C0 runs, which seem to perform better than the baseline runs, indicates that the effect of aging by mixing is either too strong or has to be compensated by other transport processes. One possible reason for a too strong isentropic mixing from the lowermost stratosphere into the tropical middle troposphere below $\zeta = 350$K could be deviation of the model levels from





|  | CO | $O_3$ | $N_2O$ | $CO_2$ |
|---|---|---|---|---|
| StratoClim | AMICA | FOZAN | HAGAR | HAGAR |
| Tropospheric $\theta$-range [K] | 320-400 | 320-400 | 320-400 | 380-400 |
| WISE | UMAQS | FAIRO | UMAQS | HAGAR-V |
| Tropospheric $\theta$-range [K] | 280-340 | 280-340 | 280-340 | 280-340 |
| lower boundary (CLaMS) | AIRS | set to 0 | CATS/NOAA | CarbonTracker |

**Table 3.** Observed species, instruments, $\theta$-ranges included into the validation procedure, and the origins of the data used for the lower boundary condition in CLaMS (AIRS - Atmospheric Infrared Sounder on NASA's Aqua satellite, Chromatograph for Atmospheric Trace Species (CATS) from NOAA, CarbonTracker from NOAA, version CT-NRT.v2017). For details of the CLaMS boundary conditions see Konopka et al. (2019) ($CO_2$) and Pommrich et al. (2014) (all other species). Tropospheric $\theta-$ranges define maximal vertical extensions where new transport modes in CLaMS-2.0 are evaluated in terms of the root mean square differences between observations and model results. The instruments and their performance during StratoClim are described in von Hobe et al. (2021) and during WISE in Krasauskas et al. (2021) and Lauther et al. (2021).

the isentropes in this part of the atmosphere. Thus, CLaMS approximation of isentropic mixing confined by the model levels
does not work correctly in this part of the atmosphere and may cause some spurious, cross-isentropic diffusion.

### 3.2 In-situ observations: CO profiles

In the following, model results are compared to in-situ data which were observed during the Geophysica campaign StratoClim
(Stratospheric and upper tropospheric processes for better climate predictions) in Nepal in July-August 2017 as well as during
the HALO campaign WISE (Wave Driven Isentropic Exchange) in Ireland in September-October 2017. While during Strato-
Clim tropical UTLS conditions within the Asian Summer Monsoon (ASM) anticyclone were sampled, WISE data represent
much more extra-tropical and mid-latitude conditions in the vicinity of the subtropical jet. All CO, $O_3$, $N_2O$ and $CO_2$ time
series measured during these two campaigns are included (all local flights) with details listed in Table 3. All time series are
transported to the closest 12 UTC synoptic time using 3d CLaMS trajectories calculated with the same reanalysis as the re-
spective CLaMS run. Tracer mixing ratios at the CLaMS air parcels closest to the observations quantify the model prediction
(no interpolation from other potential neighbors).

In Fig. 4, mean profiles of CO are plotted for each campaign and for different CLaMS setups listed in the legend. The
mean observed CO profile during StratoClim (Fig. 4a) shows typical tropical features with highest values at the surface and
around the main convective outflow between 360 and 370 K. Above this level, CO decreases, almost constantly, down to the
stratospheric background above 420 K due to its chemical lifetime of the order of a few months (von Hobe et al., 2021). The
root mean square differences calculated within the tropospheric ranges (see Table 3) quantify the performance of different
CLaMS runs and are listed in Table 4.

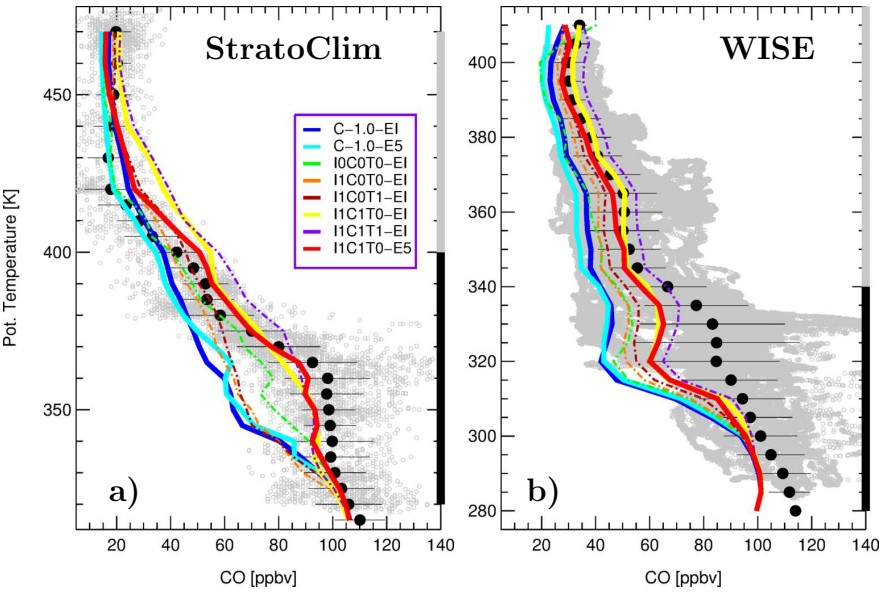

**Figure 4.** All CO observations (gray) during the StratoClim (a) and WISE (b) campaigns are shown versus CLaMS simulations. The averaged experimental data are plotted as black bold symbols (mean values are calculated by gridding all observations and using an overlapping boxcar width). The respective results for the model runs are color-coded as described in the legend. For the baseline runs (C-1.0) and for the best runs (I1C1T0) thick solid lines are used. The root mean square differences calculated between the simulations and experimental data in the tropospheric range (black solid line on the right y-axis) are listed in Table 4. The performance in the stratospheric range (gray solid line on the right y-axis) is discussed in Sect. 3.6. Note the different y-axis ranges in a/b.

A clear improvement in the model representation of the main convective outflow is achieved when unresolved convection is switched on (C1) and, consequently, the observed CO values around 90 ppbv between 360 and 370 K are reproduced by the model. As discussed in von Hobe et al. (2021), photolytic decay of CO in slowly ascending air within a well-confined ASM anticyclone defines the vertical gradient of CO between 365 and 420 K. Above this level, mixing with the surrounding stratosphere determines the CO mixing ratios. The vertical gradient of CO between 400 and 450 K is better represented in E5- than in EI-driven simulations (I1C1T0-E5 versus I1C1T0-EI). Tropospheric mixing (from I1C1T0 to I1C1T1) shifts CLaMS profile to too high values suggesting that this transport enhances the numerical diffusion rather than improves our simulations. Fig. 4b shows mean CO profiles during WISE when mainly mid-latitude air was sampled. Comparison with CLaMS results confirms the importance of convection parameterization although CO values between 310 and 340 K are still underestimated. Remarkable is the CO minimum around 315 K which is much more pronounced in CLaMS simulations than in the observations. This difference between model and observations can either be explained by too much downward transport or not enough convective updrafts in the model.

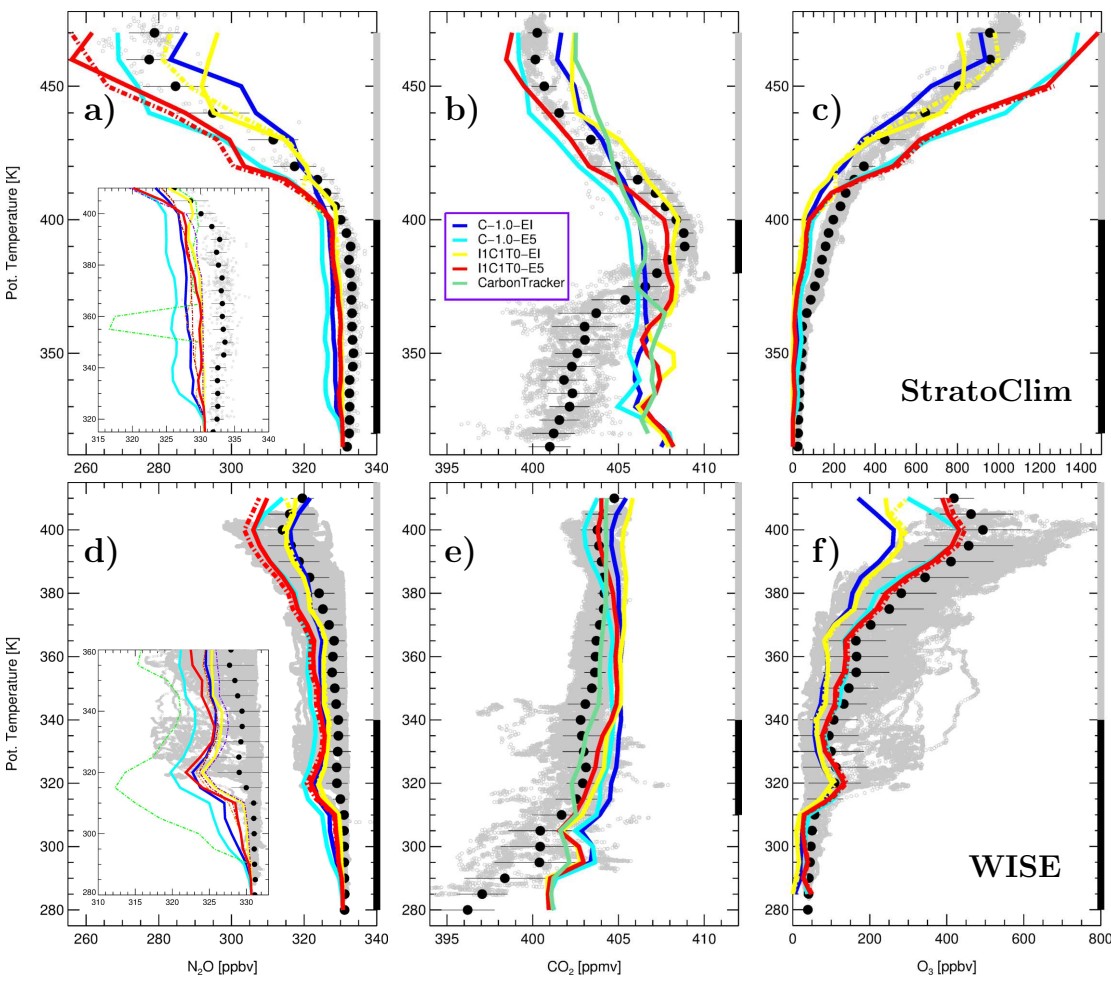

**Figure 5.** Same as in Fig. 4 but for $N_2O$ (a,d), $CO_2$ (b,e) and $O_3$ (c,f) observations versus CLaMS for the most important model configurations listed in the legend of Fig. 4. For the baseline runs (C-1.0) and for the best runs (I1C1T0) thick solid lines are used. Dashed thick lines denote results of the respective perpetuum runs. Top and bottom panels show the StratoClim and WISE related profiles, respectively. $CO_2$ profiles directly obtained from the CarbonTracker data are also shown (beige green). In the subpanels for $N_2O$, details of tropospheric profiles are higher resolved and color-coded as in Fig. 4. Table 4 compares the root mean square differences and Sect. 3.6 discusses the stratospheric performance within the tropospheric and stratospheric ranges, respectively.

## 3.3 $N_2O$, $CO_2$ and $O_3$ profiles

Now, we compare in Fig. 5 the same type of mean profiles like for CO (Fig. 4) but for $N_2O$ and $CO_2$ which are tropospheric tracers with much longer lifetimes than CO (~hundred years for $N_2O$ and even centuries for $CO_2$). Below the tropopause,
$N_2O$ does not show much variability and can be characterized by a weak positive and negative mean vertical gradient for the





StratoClim and WISE data, respectively (Fig. 5a, d). The deviation of the simulated from the observed gradients measures the quality of the CLaMS runs; for a more detailed, zoomed view see the subpanels of Fig 5a and Fig 5d. Note that the pure advection run (I0C0T0) shows the worst performance with few very low (unmixed) stratospheric values in the lower troposphere. The best performance can be diagnosed for the I1C1 or even I1C1T1-runs. Note also a layer around $\theta = 320$ K in the extratropical WISE data (Fig- 5d) with low $N_2O$ values marking signatures of stratospheric intrusions. This layer is much more pronounced in CLaMS simulations than in the observations and is consistent with a similar feature in CO profile (Fig. 4b).

Mean $CO_2$ profiles (Fig. 5b, e), can be understood as a result of a coupled tropospheric and stratospheric transport folded in time and space with the evolution of the $CO_2$ sources (and sinks) in the PBL while both, transport and sources show a pronounced seasonality (Diallo et al., 2017). $CO_2$ being almost chemically inert is a well suited tracer for validation of transport in the models if the lower boundary condition is realistically parameterized. However, the (assimilated) CarbonTracker data set used for the lower boundary of CLaMS does not reproduce well the $CO_2$ observations in the lowest part of the troposphere where air dominated by regional sources was sampled (Kathmandu/Nepal for StratoClim and Shannon/Ireland for WISE). These regional conditions which are not well resolved in the CarbonTracker data (thick green line in Fig. 5b, e) are becoming less important around the tropopause where global rather than regional source regions contribute to the observed $CO_2$ mixing ratios. Ray et al. (2021) have shown that during the summer monsoon season over North America, the observed $CO_2$ profiles above 380 K are influenced rather by tropical than regional sources and even dominated by such sources above 420 K.

Thus, we compare now the observed mean profiles of $CO_2$ with CLaMS simulations only above $\theta$ larger than 380 and 310 K for StratoClim and WISE observations, respectively. First, the upward propagation of the $CO_2$ seasonal cycle during StratoClim is better reproduced in the updated CLaMS-2.0 version including parameterization of unresolved convective updrafts and improved boundary conditions (I1C1T0-EI/E5) compared to the former model version CLaMS-1.0 (C-1.0-EI/E5) and even better than by the CarbonTracker data set as can be deduced from the comparison around $\theta = 380$ K. Second, above this level, the upward propagation of the $CO_2$ signal is rather too fast in the EI driven simulation and too slow for the E5 meteorology in some agreement with our CO-related results. The same type of behavior can also be diagnosed during WISE with a better performance of CLaMS-2.0 in the tropospheric regime between 310 and 340 K. Note that one year simulations (2017) limit the interpretation of the upward propagation of the $CO_2$ signal and results of the perpetuum run cannot be applied because of a too strong positive $CO_2$ trend in the PBL.

Finally, $O_3$ profiles are shown in Figs. 5c, f. In the mid-latitudes during WISE, $O_3$ gradients are better reproduced in the E5-driven runs while mean profiles of $N_2O$ are better represented by the EI-driven simulations. Because only a simplified $O_3$-chemistry is used (Pommrich et al., 2014) and the upper boundary of $O_3$ is defined at $\theta = 500$ K by HALOE climatology, CLaMS results should be interpreted with caution. Especially, some missing chemical loss- or production cycles of $O_3$ may be the reason for a wrong interpretation of the vertical gradients of ozone. The low bias of $O_3$ below 410 K during StratoClim is only partially related to our lower boundary condition setting $O_3$ to zero and more likely caused by the pollution-induced ozone production in the monsoon anticyclone which is not represented in the model.

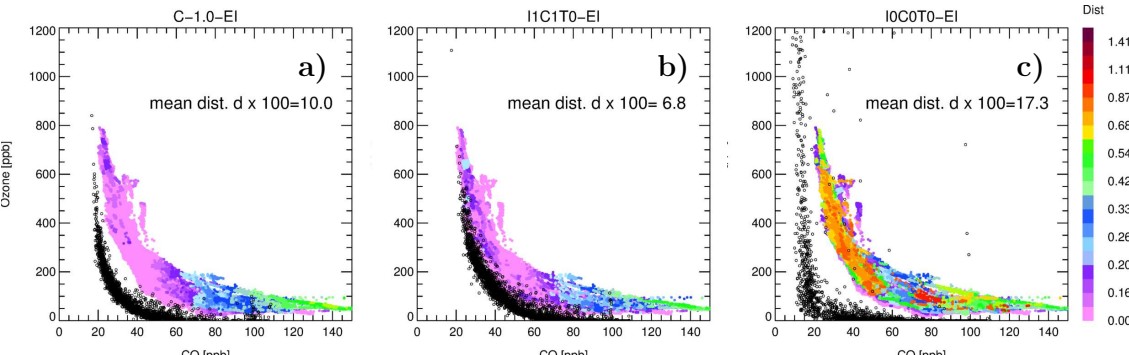

**Figure 6.** CO-$O_3$ correlation-based metrics. Observed (colored) versus simulated (black) CO-$O_3$ correlation for all flights of the WISE campaign. The observed data are color-coded with the mean (Euclidean) distance $d$ calculated in the CO-$O_3$ space after normalizing CO and $O_3$ data by 200 ppbv and 3000 ppbv, respectively. With $d_i^2 = (x_i^o - x_i^s)^2 + (y_i^o - y_i^s)^2$ denoting the square of the Euclidean distance between the observed ($o$) and simulated ($s$) (normalized) CO/$O_3$ ($x/y$) value pairs $i$; $d$ is defined as the arithmetic mean over all $d_i$ values. Note that $d$ is a dimensionless quantity varying between 0 and $\sqrt{2}$ (here multiplied with 100). (a) baseline run (C-1.0-EI), (b) isentropic mixing and unresolved convection (I1C1T0), (c) pure advection (I0C0T0).

### 3.4 CO-$O_3$ correlation

A convenient diagnostic of transport in the UTLS offers the CO-$O_3$ correlation observed in the UTLS (Hoor et al., 2002; Pan et al., 2004; Kunz et al., 2009). In the CO-$O_3$ space, this correlation is characterized by CO and $O_3$ branches defining the (mean) troposphere and stratosphere, respectively, and by the remaining part sampled in the UTLS region (Fig. 6). In
the extratropical UTLS, this part is formed mainly by isentropic mixing in the vicinity of the jets (Konopka and Pan, 2012) while in the Tropical Tropopause Layer (TTL) photochemistry rather than isentropic mixing is the reason (Vogel et al., 2011; von Hobe et al., 2021). To focus on mixing processes close to the subtropical jet, we include only WISE data in the following analysis.

A comparison between the observed and simulated CO-$O_3$-correlation is shown in Fig. 6, exemplarily for three EI-driven
CLaMS configurations: CLaMS-1.0 baseline run (Fig. 6a), CLaMS-2.0 including isentropic mixing and unresolved convective updrafts (Fig. 6b), and for the pure advection run (Fig. 6c). In a perfect model, the observed and simulated correlations should be equal. Deviations from such an idealization can be used to evaluate the model transport representation. Here, following the procedure described in Konopka et al. (2004) to optimize the CH$_4$-H1211 correlation, a mean (Euclidean) distance $d$ calculated in the normalized CO-$O_3$ space is used to measure such deviations (for exact definition see caption of Fig. 6).

To reconstruct correctly the observed CO-$O_3$ correlation both the end members of the mixing lines and the isentropic mixing itself should be well-represented in the model. Especially the CO values of the air parcels undergoing isentropic mixing in the model are preconditioned by the representation of convection in the model. In this way, the CO-$O_3$ correlation based metric rates not only isentropic mixing but also the quality of the other transport modes like parameterized unresolved convection
or tropospheric mixing. As discussed in Pan et al. (2006), missing isentropic mixing in pure advective studies explains a poor



| | $\Delta CO$ [ppb] | | $\Delta N_2O$ [ppb] | | $\Delta CO_2$ [ppm] | | $d(CO\text{-}O_3)\times 100$ | | $r$ |
|---|---|---|---|---|---|---|---|---|---|
| | StratoClim | WISE | StratoClim | WISE | StratoClim | WISE | StratoClim | WISE | |
| C-1.0-EI | 29.1 | 40.6 | 5.1 | 4.9 | 2.1 | 2.4 | 11.0 | 10.0 | 0.54 |
| I0C0T0-EI | 23.1 | 40.4 | 16.5 | 36.7 | 4.4 | 4.16 | 13.7 | 17.3 | 1.0 |
| I1C0T0/T1-EI | 25.1/22.8 | 36.0/30.8 | 4.7/4.4 | 5.1/4.4 | 1.4/1.2 | 2.5/2.0 | 9.7/9.4 | 8.5/7.6 | 0.49/0.43 |
| I1C1T0/T1-EI | 15.8/16.3 | 27.9/21.9 | 3.5/3.2 | 4.6/3.7 | 1.3/0.8 | 2.0/1.3 | 7.3/7.5 | 6.8/6.6 | 0.39/0.34 |
| I1C1T0-E5 | 15.2 | 27.3 | 4.2 | 6.6 | 1.2 | 2.0 | 8.8 | 6.2 | 0.80 |
| I1C1(30K)T0-E5 | 16.8 | 31.7 | 4.9 | 6.9 | 1.3 | 1.9 | 8.4 | 6.7 | 0.83 |
| I1C1(-1.0)T0-E5 | 17.5 | 30.4 | 5.1 | 6.7 | 1.6 | 1.9 | 8.9 | 6.6 | 0.90 |
| I1C0T0-E5 | 24.4 | 36.9 | 6.9 | 7.5 | 1.9 | 2.2 | 19.8 | 8.3 | 1.0 |

**Table 4.** Rating of the EI and E5-driven runs (upper and lower parts of the table) in terms of the root mean square differences $\Delta$ calculated over the vertical ranges listed in Table 3 and in terms of the mean deviations $d$ from the observed CO-$O_3$ correlation during StratoClim and WISE (smallest values of $\Delta$ an $d$ indicate the best performance). These four metrics $m_i$ are normalized by their maximal values within each of the considered group and are used to calculate the cumulative rating parameter $r$ defined as a sum of all metrics $m_i$ normalized by the sum maximum (i.e. $r = 1$ mean the worst case and smallest value of $r$ means the best case). While for the first four cases (EI runs) $r$ is calculated relative to the I0C0T0-EI, $r$-value of the last four cases (E5 runs) are determined relative to I1C0T0-E5.

representation of the observed CO-$O_3$ correlation (Fig. 6c). Thus, CLaMS-1.0 configuration (Fig. 6a) including isentropic mixing performers better, with smaller $d$ values as in Fig. 6c. The CLaMS-2.0 run (I1C1T0, Fig. 6b) gives in this example the smallest $d$ value, i.e. the best representation of the observed CO-$O_3$ correlation, by combining uresolved convective drafts with isentropic mixing.

**3.5  Rating of different transport scenarios**

Table 4 summarizes the rating of all runs, both in terms of the root mean square differences and by using the parameter $d$ discussed above. With exception of CO, pure advection runs show by far the worst performance. For CO, CLaMS-1.0 is the worst case mainly because the update frequency of the lower boundary condition is lower in CLaMS-1.0 than in the pure advection run (24h versus 6h, see also discussion related to the Fig. 2, a slightly thinner lower boundary compared with CLaMS-

15 2.0 runs plays only a minor role). To compare all EI-related results (upper part of Table 4), the cumulative rating parameter $r$ was determined (for exact definition see caption of Table 4) with $r = 1$ quantifying the worst case (pure advection) and with the smallest values of $r$ pointing to our best cases (last column in Table 4). We conclude that the largest improvement was achieved by including isentropic mixing in the model simulation (I1 versus pure advection I0). The second largest improvement results by adding to isentropic mixing the parameterization of the unresolved convective uplifts. By including tropospheric mixing (numbers behind the slash) only weak improvement was found. Note that the best E5-driven run (I1C1T0-E5) performs slightly better than the respective EI simulation (I1C1T0-EI) in terms of $\Delta CO$, $\Delta CO_2$ and $d$ (WISE) but not in terms of $\Delta N_2O$.



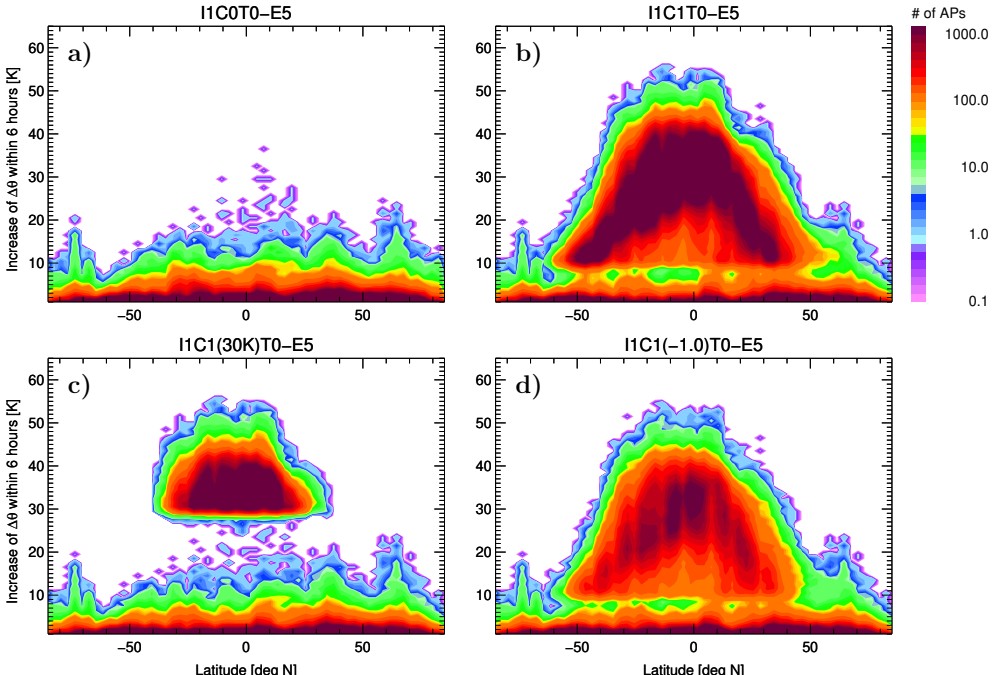

**Figure 7.** Total number of APs lifted upwards (units: total number per lat/theta grid size and time step) in different E5-driven configurations averaged over the first 10 days of January 2017. (a) Run without any convective parameterization. (b) Run with the parametrization of unresolved convection and with the best agreement with experimental data. (c and d) Two sensitivity studies for unresolved convection with $\Delta\theta_c > 30$ K (instead of $\Delta\theta_c > 10$ K, left) and for $N_m^2 = -1.0 \times 10^{-4}$ 1/s$^2$ (instead of $N_m^2 = 0$ 1/s$^2$, right).

Finally, to further illustrate the contribution of unresolved convective uplifts to transport in the troposphere, we analyse differences in the vertical distribution of air parcels in CLaMS for configurations with different strengths of parameterized convection. For this reason, forward E5-driven trajectories are globally started, every 6 hours, from the CLaMS PBL uniformly covered by about 68000 air parcels (APs). The time length of every trajectory is also 6 hours. Positions at the start and at the end of every trajectory are used to calculate $\Delta\theta$ of air parcels lifted upwards. The first 10 days (i.e. almost 40 time steps) of January 2017 are considered. We compare the run I1C1T0-E5 (which performs similarly well as the I1C1T0-EI run) with runs where the convection parametrization was reduced (I1C1(30K)T0, I1C1(-1.0)T0-E5) or even completely switched off (I1C0T0).

Fig. 7 shows such a comparison using the total number of the uplifted air parcels as a function of latitude and the uplift $\Delta\theta$ averaged over all (advective) time steps while Table 4 (lower part) compares these runs using metrics discussed above. Both sensitivity runs perform worse compared with the best case. Starting from a pure isentropic run I1C0T0 as a reference with the cumulative metrics equal 1 (worst case), the improvement of the runs with convection can be quantified by 0.90, 0.83 0.80 for the I1C1(-1.0)T0, I1C1(30K)T0 and the best run I1C1T0, respectively The comparison with the run without any parameterization of convection (Fig. 7a) shows that a massive redistribution of APs is necessary (from d, c, to b) in order to reproduce the observed features like the main convective outflow in the CO/N$_2$O profiles.

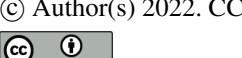


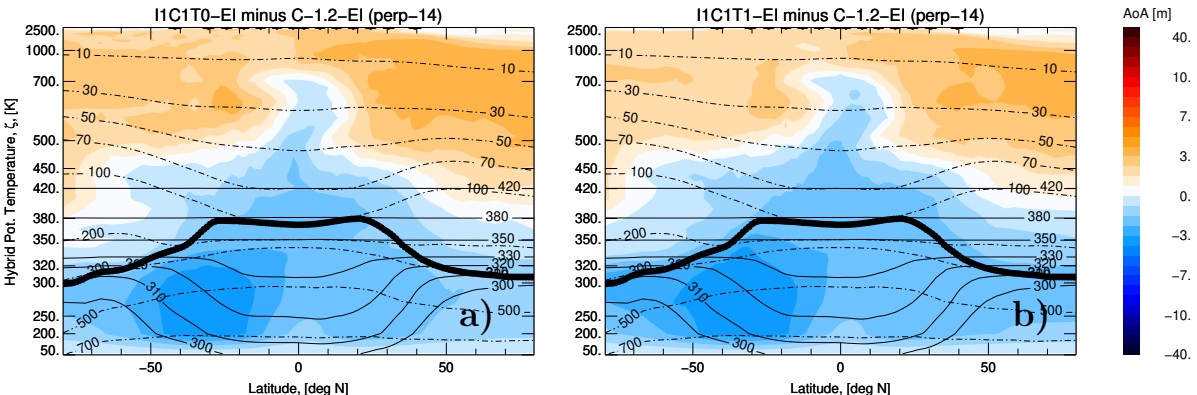

**Figure 8.** Impact of tropospheric transport modes on the distribution of the age of air AoA (in months) shown as a difference to the CLaMS-1.2 AoA distribution (December 2017, perpetuum 14-years): with unresolved convection (a) with unresolved convection and tropospheric mixing (b). The results for the respective E5-driven simulations are very similar in the troposphere; the stratospheric effect of aging by mixing is even weaker (not shown).

### 3.6 Stratospheric performance

Because CLaMS-2.0 runs cover only one year, the impact of the new tropospheric transport modes on the composition of tracers in the stratosphere is not well-resolved. For such studies, transient runs of the order of 10 years would be necessary to correctly reproduce the old air masses. Here, CLaMS-2.0 perpetuum runs for 2017 are used to mimic such transient runs. Furthermore, the simulated tracer distributions in the stratosphere are mainly determined by the representation of the Brewer-Dobson circulation (BDC) in the reanalyses and are well-documented for the EI (Ploeger et al., 2019) and E5-driven (Ploeger et al., 2021) multi-year CLaMS-1.0 simulations. While EI-driven runs show a too fast BDC, with too young air masses in the tropical lower stratosphere, the reverse effect was diagnosed for the E5-driven runs with rather too slow BDC and too old air in the TTL. These findings are consistent with stratospheric parts of the simulated $O_3$ and $N_2O$ profiles shown in Fig. 5 (thick dashed lines for the perpetuum runs). Thus, in the tropical stratosphere above 400 K during StratoClim, E5-driven runs seem to produce too strong stratospheric gradients of $O_3$ and $N_2O$ while the EI-based simulations compare better with the observations.

We discuss now how the tropospheric modes of transport implemented in CLaMS-2.0 influence the distribution of the age of air (AoA). Fig.8 shows that tropospheric air becomes younger for runs with unresolved convection (Fig. 8a) and even younger if tropospheric mixing is added (Fig. 8b). The strongest effect is diagnosed in the middle troposphere around 40S and reflects the seasonality of convection shown here for December 2017. The rejuvenation of air propagates upwards into the tropical pipe and covers the whole loewermost stratosphere below $\theta = 380$ K. Above this level, re-circulation and aging by mixing (Garny et al., 2014) makes the air older due to slightly changed intensity of isentropic mixing in CLaMS-2.0 compared with CLaMS 1.0, although the relative values of these changes are small.



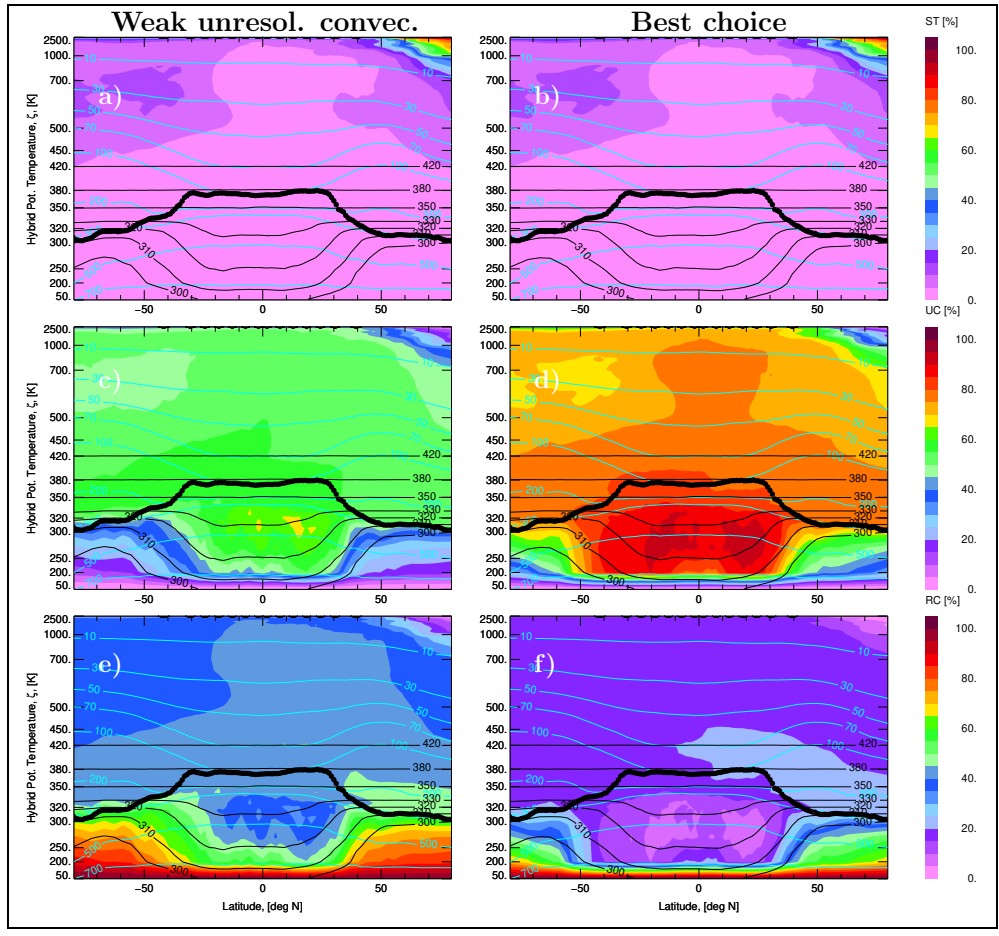

**Figure 9.** Budget of three origin tracers quantifying the contributions of the upper boundary (ST, panels (a) and (b)) and of the lower boundary of CLaMS (UC, RC shown in panels (c), (d) and (e), (f), respectively). ST, UC and RC are calculated for the last day of the 14-years perpetuum run (31.12.2017) for the weak contribution of the unresolved convection (left column, run I1C1(-1.0/30K)T0-E5) and for the best case (right column, run I1C1T0-E5). By repeating 14-times year 2017, a steady state with respect to these tracers was reached, i.e. ST+UC+RC=100%. With lower boundary approximating the PBL, UC quantifies the contribution of unresolved convective updrafts and RC the contribution of the PBL following the transport resolved in the ERA5 reanalysis.

The unresolved convection strongly influences the pathways of air into the stratosphere. For further illustration, idealized tracers denoted as ST and TR are released in the upper and lower boundary of CLaMS, respectively. The zonal mean of their spatial distribution is calculated for the last day of the 14 years perpetuum run (31.12.2017) and shown in Fig. 9. In addition, 5 TR is divided into two subsets, UC quantifying the contribution of the unresolved convective updrafts and RC measuring the amount of air which does not experience any parameterized updrafts. All three tracers add up to 100% and quantify the amount of air following different pathways: from the top of the stratosphere (ST), from the Earth's surface via the unresolved convective



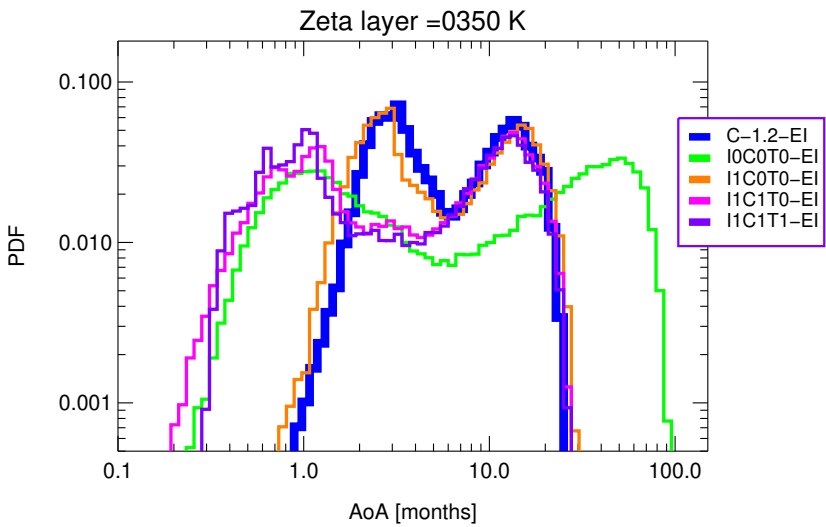

**Figure 10.** PDF of age of air (AoA) after 14 years of perpetuum runs averaged over all APs within the CLaMS layer around $\zeta = 350$ K

updrafts (UC) and from the other Earth's sources following the transport resolved in ERA5 (RC). Note that by switching off the convection parameterization, UC=0 is valid. As can be seen in the left column of Fig. 9, even a weak contribution of convection
10   parameterization leads to UC values around 50% between the tropopause and $\sim 50$ hPa.

However, in order to reproduce the observed tropical main convective outflow like that diagnosed in the StratoClim CO observations, much stronger contribution of unresolved convective updrafts is necessary (best case, second column in Fig. 9), indicating UC values around 75% in the lower stratosphere. We conclude, that it is very unlikely to transport air from the boundary layer into the stratosphere following only the pathway with convection resolved by the reanalyses. Using e90-tracer
15   based diagnostics, Hoffmann et al. (2022) have shown that even the highest available spatial and temporal resolution of E5 meteorological data (i.e. time resolution=1 hour, spatial resolution=0.3 deg) does not significantly enhance the contribution of the resolved transport.

Finally, to visualize how much younger the air in the UTLS region is if tropospheric modes of transport are switched on, the PDFs of AoA are compared in Fig. 10 for different EI-driven model runs. These PDFs are calculated for all air parcels within the CLaMS layer around $\zeta = 350$ K for the last day (31.12.2017) of the respective perpetuum run. As intended, there is no
significant difference between the pure isentropic CLaMS-2.0 run (I1C0T0) and the reference run C-1.2. Including unresolved convection (C1T0) or unresolved convection and tropospheric mixing (C1T1) significantly enhances the contribution of young air to the air composition around $\zeta = 350$ K. This is even slightly more than pure advection studies can provide. In addition, pure advection transport produces very old air which, as shown in the previous section, does not match experimental data.





## 4  Conclusions

Including unresolved tropospheric transport processes like convection and mixing, as implemented in the newly released CLaMS model version (CLaMS-2.0), improves tracer representation of the UTLS in the model. While the parameterization of unresolved convection enables reproducing the signatures of the main convective outflow as observed in the tropical CO profiles, the potential improvements related to the parameterization of tropospheric mixing in CLaMS-2.0 are less obvious. As deduced from the in-situ based metrics, tropospheric mixing seems to remove some tropospheric variability in the simulated

$N_2O$ profiles and to homogenize vertically such profiles, in some agreement with the observations. On the other hand, the e90-based tropopause is slightly too high in the tropics and the comparison with the CO profiles within the ASM anticyclone indicates also a too strong contribution of tropospheric mixing.

Although both parameterizations, i.e. those of unresolved convection and of tropospheric mixing only weakly influence the stratospheric distribution of mean age of air, their influnce on the young part of the age spectrum and therfore on the very

short-lived substances (VSLS) like the ozone-depleting, halogen-containing substances is still not quantified. Especially the parameterization of unresolved convection strongly influences the potential pathways of transport from the Earth's surface not only into the UTLS region but also into the whole stratosphere. The dominant contribution of the convective pathway, i.e. from the boundary layer via the main convective outflow into the stratosphere (see Fig. 1), to the composition of stratospheric air highlights the importance of including a convection parametrization in global models which are not resolving convective up-

and downdrafts.

## Appendix A:  Vertical grid and adaptive grid procedure

The Lagrangian (irregular) grid of air parcels (APs) covers the whole atmosphere from the surface to around 50 km or $\zeta = 2500$ K with $\zeta$ being the hybrid potential temperature (Mahowald et al., 2002). The initial positions of APs are generated following the concept described in Konopka et al. (2012) (i.e. by assuming that the aspect ratio is controlled by the static

stability and that the entropy of the system is uniformly distributed over all APs). However, this concept, mainly developed for the free atmosphere, may be not valid in the vicinity of the Earth's surface where atmospheric transport is dominated by the boundary effects like friction and convection. Thus, we decouple the horizontal and vertical resolution of the lowest model layer (approximating the planetary boundary layer (PBL)) from the grid generation procedure in the sense that both the vertical and horizontal separation of the APs in this layer can be freely chosen (Fig. A1).

Isentropic mixing of APs in CLaMS is based on the adaptive grid procedure with interpolations triggered by flow deforma-tions detected layer-wise during each advective time step $\Delta t$. The APs covering the whole atmosphere are divided into layers which are parallel to isentropes roughly above $p_r = 300$ hPa. The model layers directly above the Earth's surface follow the

5  orography rather than isentropes. In every layer, the adaptive grid procedure inserts and merges APs in the regions of strong flow deformations. Let $r_0$ be the mean separation between the APs in an arbitrary chosen layer. Then, two critical distances, $r_\pm$ are defined as $r_\pm = r_0 \exp \pm \lambda \Delta t$, with the Lyapunov exponent $\lambda$ parameterizing critical deformation triggering adaptive interpolations after inserting ($r > r_+$) or after merging ($r < r_-$) of the APs.

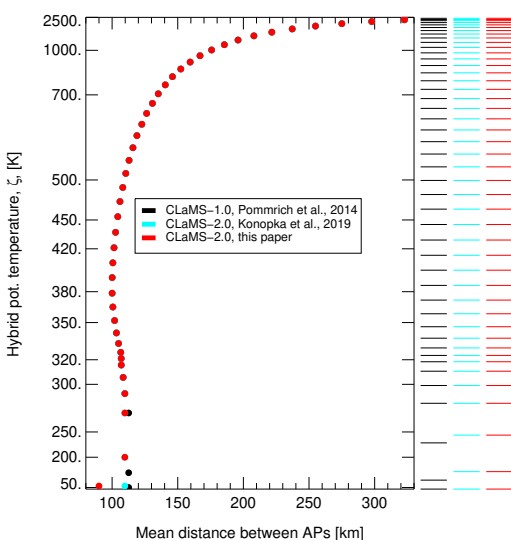

**Figure A1.** Initial distribution of APs with slightly enhanced thickness of the lowest layer (from $\zeta = 100$ to 140 K, i.e. from CLaMS-1.0 to ClaMS-2.0) and with slightly increased horizontal resolution in this layer used here ($r = 90$ km instead of $r = 110$ km) for grid configuration generated with $r_0 = 100$ km and aspect ratio $\alpha = 250$ at $\zeta = 380$ K following procedure described in Konopka et al. (2012). For the here used EI/E5 reanalyses, the zonal horizontal resolution varies between 111 km at the equator and 78 km at 45 degree latitude. With the choice $r = 90$ in the lowest layer, we aim to include better the resolved tropical convective updrafts.

In general, the adaptive grid procedure increases the total number of APs up to 50% relative to the initialization (in particular layers this number is even higher). Especially in the deformation-dominated zones, like outer flanks of the jets (like polar or subtropical jets) or in the tropical stratosphere within the QBO wind reversal, a temporal increase of APs number can be diagnosed. Thus, the merging step of the adaptive grid procedure balances the deformation-induced insertion of new APs and keeps the total number of APs within a certain range (Fig. A2).

In CLaMS-2.0 shorter advective time steps $\Delta t$ are applied in order to resolve the diurnal cycle of unresolved convection (6 instead of 24 hours in CLaMS-1.0). For a better control of the adaptive grid procedure, we redefined $r_-$ to $r_- = \epsilon r_0 \exp(-\lambda \Delta t)$, with $\epsilon$ regulating the intensity of merging keeping the total increase of APs (relative to the initial value) below around 50%. The use of the parameter $\epsilon$ is new. Typical values are around 0.30 for $\Delta t = 6$ hours and $\lambda = 4.0$ day$^{-1}$ although other choices are possible and need further investigation (Fig. A3). The choice $\epsilon = 1$ with $\Delta t = 24$ hours restores CLaMS-1.0 configuration. Parametrization of convection within CLaMS-1.0 configuration works only with $\Delta t = 24$ because convection and mixing frequency must be equal in order to follow the relative motion of all advected APs.



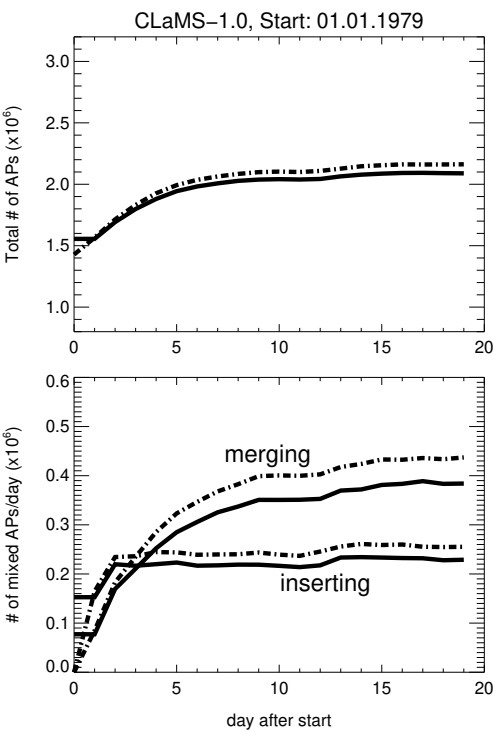

**Figure A2.** Starting with the initial distribution of APs (01.01.1979, baseline run), their total number increases in the CLaMS-1.0 standard configuration during the first 2 weeks of the simulation for both EI (solid) and E5 (dashed) cases (top panel). The bottom panels shows the numbers of APs after mixing step due to insertion (lower branch) and due to merging (upper branch). A steady state is reached after about 10 days. Note that the number of merged APs is always larger than the number of inserted APs. This is because a large part of inserted APs is subsequently merged and, consequently, counted as merged and not as inserted.

The need of using $\epsilon < 1$ is obvious for $\Delta t = 6$ hours. For such configurations, the elimination step (with $\epsilon = 1$) is too strong compared with the insertion step. Consequently, the total number of APs decreases from the initial configuration by more than 50% (Fig. A3, red line). Although it is possible to start the model with respectively larger number of APs, it is advantageous to enhance the number of APs in strongly deformed parts of the flow where tracer filaments and small-scale structures are very likely and where higher density of APs can resolve better their structures. Using the parameter $\epsilon$ it is possible to tune the model in order to increase the model resolution in the mixing zones.

*Author contributions.* PK, FP, MT and MvH conceived the presented ideas. PK performed the numerical simulations and wrote the paper. LH contributed with his experience related to convection parameterization and to the high-resolution ERA5 data. CK, MvH, FR, CMV, VL, AZ, PH provided and helped to use the experimental data obtained during the StratoClim and WISE campaigns. All authors contributed to finalizing the paper.

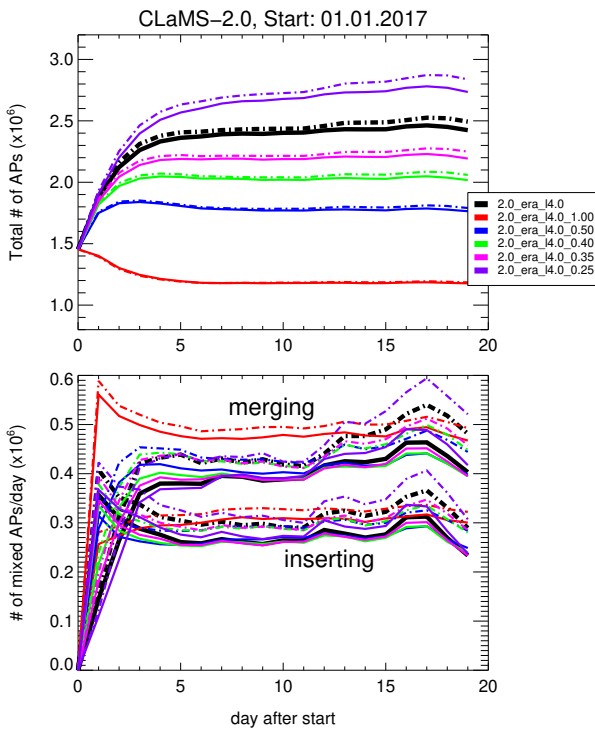

**Figure A3.** Tuning of $\epsilon$ for runs with $\Delta t = 6$ hours, $\lambda = 4.0$ day$^{-1}$. Here different values of $\epsilon$ are used, from $\epsilon = 1$ (CLaMS-1.0) to $\epsilon = 0.25$. To get the total number of APs which is similar to that in CLaMS-1.0 reference run, $\epsilon = 0.3$ is recommended (black). Solid and dashed lines denote the EI and E5-driven runs, respectively.

*Acknowledgements.* The European Centre for Medium-Range Weather Forecasts (ECMWF) provided meteorological analysis for this study. We thank Alexey Ulanowski and Vladimir Yushkov for their support related to the O$_3$ FOZAN data. Excellent programming support was provided by N. Thomas. The authors gratefully acknowledge the project CLaMS-ESM of the Earth System Modelling Project (ESM) for funding this work by providing computing time on the ESM partition of the supercomputer JUWELS at the Jülich Supercomputing Centre (JSC). Jens-Uwe Grooß and Gebhard Günther strongly contributed to the data management on the supercomputer. We also thank Johannes Wintel and Thorben Beckert who supported HAGAR operations and data analysis during StratoClim, as well Johannes Wintel, Andrea Rau and Emil Gerhardt for their support with the HAGAR-V instrument during WISE. We also acknowledge Nicole Spelten for preparing time-synchronized merged files that were used for our analyses.

*Code and data availability.* CLaMS-2.0/MESSy is available as a part of the Modular Earth Submodel System (MESSy), Version 2.54 at the Mercurial server http://messy.fz-juelich.de/messy-2.54.0-clams. ERA5 and ERA-Interim model level reanalysis data are available from the ECMWF as deterministic forecasts (atmospheric model): ERA5 via: https://apps.ecmwf.int/data-catalogues/era5/?class=ea, ERA-Interim via: https://apps.ecmwf.int/archive-catalogue/?class=ei). The CarbonTracker data (version CT-NRT.v2017) can be downloaded from the



NOAA ftp server: ftp.cmdl.noaa.gov, see: /products/carbontracker/co2/CT-NRT.v2017/molefractions/co2_total/. The StratoClim data can be downloaded from: http://www.stratoclim.org/ and the WISE data from: https://halo-db.pa.op.dlr.de/. For more detailed model data, please contact the authors.





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
