# Peer review of "Tropospheric transport and unresolved convection: numerical experiments with CLaMS-2.0/MESSy"

_Geoscientific Model Development, 2022_

## Author Comment (AC1)

Response to reviewer 1

We would like to thank reviewer 1 for a very thoughtful and detailed review of our manuscript that helped to improve the paper. Both reviewers criticized too "technical" notation. Because of this, we improved our notation, simplified Table 2 and tried to avoid techical abbrevations in the main text. Furthermore, a too long chapter 3 was shortened and a new chapter "Discussion" was included. In the last 3 paragraphs of this new chapter, we discuss many of the points raised by both reviewers. In the following, we address all other points (denoted by italic letters). We reference to the changes in the manuscript (marked with red color and attached to this reply) by giving the page number (PXX).

Major comments:

1. *This paper describes a new version 2.0 of the CLAMS-MESSY chemistry-transport model. The model is Lagrangian and follows isolated air parcels (APs) around the atmosphere, being required every so often to remap the parcels because there will be regions with a glut, and regions deserted by the APs. The real art here is remapping onto a std grid or figuring out how to remove and add parcels. The new model presented here includes inter-Parcel exchange via "parameterization of tropospheric mixing and unresolved convection." What is worrisome here is that there is no clear atmospheric physics to determine the rate of mixing, but rather it is just tuned and is defined to represent a type of mixing. It appears that CLaMS-2.0 has already been published and this paper is an application of it. If so, is this a GMD paper or an ACP one? I am not too worried about which, but the editor may wish to weigh in. Overall, this is a reasonably nice paper, written clearly and deserves to be published after some thought and revision. I include comments by line number below.*

   Yes, we agree that this paper is an application of CLaMS-2.0 presented in Konopka et al., GMD, 2019. However, there are indeed several important technical updates and innovations related to the mixing scheme and to implementation of the model within the Modular Earth Submodel System (MESSy). Thus, we think that GMD instead of ACP is a better platform for this paper. The connection of our mixing scheme to the physics is difficult to validate as mixing itself is a complicated multi-scale process, both in time and space. It is more the climatological properties of mixing (i.e. statistical properties of mixing) and not singular mixing events that are well-reproduced in our model. In the new version of the manuscript we included few new sentences related to the physical motivation and physical background of our mixing parameterization. It should simplify to read the paper but it does not make completely dispensable to look in our previous publications.

2. PLEASE use continuous number, discerning page number as well as line numbers is not nice. I cannot see page number when reading your paper on a screen. Thus I may not get the page numbers correct in my comments

   Unfortunately, the continuous numbering of lines is not provided by the Copernicus Latex template. In the revised version we use red color to mark our changes and page numbers (PXX) to reference such changes in this reply.

Minor comments:

- *L15 "The second most important transport process considered is unresolved convection." I remain very confused by this term: if the convection is not resolved in the EC fields (which it is resolved explicitly in the fields I use from the IFS system) then how can CLAMS use it? Just make up a convective rate?*

  To drive the advective part of transport in CLaMS, we use the velocity fields provided by the reanalysis, i.e. $\Omega := dp/dt$ and temperature tendencies from which $\dot{\theta}$ is derived. The IFS of each reanalysis has its own convection parameterization, e.g. to drive the hydrological cycle, in particular to transport moisture within the convective cells in the model. While $\Omega := dp/dt$ and $\dot{\theta}$ define the resolved part of convection, some of the stored parameters of the convection parameterization (e.g. de- and entrainment rates) could be used to "reconstruct" the contribution of the convection parameterization on transport. Our convection parameterization is independent from such reanalysis-dependent approach. We tried to clarify the related description on P15-16.

- *P2L5 "reduced numerical diffusion compared to the Eulerian-based transport models" I am tired of Lagrangrian models mantra that Eulerian are more diffusive. Some tracer transport Eulerian schemes have negligible diffusion - please see the Lauritzen papers (Geosci. Model Dev., 7, 105-145, doi:10.5194/gmd-7-105-2014 and Geosci. Model Dev., 5, 887-901, doi:10.5194/gmd-5-887-2012. Both methods have their advantages, but you need to retract the old arguments.*

  Yes, we agree. We reformulated the corresponding sentences. See P2

- *L13: " perpetuum runs are performed (14-times 2017) as "This is a very bad approach if you are serious looking at the strat-trop region, because at Jan 1 there is a huge discontinuity in the tropopause and jet stream every annual cycle. This creates havoc with lots of instant strat-2-trop placement of ozone in the troposphere (and vice versa)."*

  Yes, we agree. Two new sentences explaining the advantages and disadvantages of the perpetuum runs is now included. See P3

- *L18: "three parameterized components of transport: isentropic mixing (I), unresolved convection (C) and tropospheric mixing (T) schematically" If these are all parameterized and not based on atmospheric physics then I do not see how you are running ERA fields. There is only a certain amount of such mixing that is consistent with the wind fields. I fear that your model is inconsistent with the ERA model result.*

  The resolved advective part of transport is just consistent with ERA, as it is driven by the reanalysis wind/heating rate fields. For the unresolved part of transport it is, in general, difficult to separate the unresolved physics (like mixing or convection) in the background reanalysis from that in our transport scheme. However, we discuss some aspects of such differences in the last 3 paragraphs of the section "Discussion". See P17-19

- *P4. The table shows a worrying feature. You have only the instant winds every 6 hours. You really need 3-hour fields if you are going to resolve any diurnal cycles, such as BL mixing. With 6h, you alias all these cycles at different points as you cross longitudes. It seems like your mixing parameters are arbitrarily selected and not related to the local meteorology. I cannot understand it.*

We agree that the time resolution is a critical issue here. However, we are constrained by the time resolution of the reanalysis. Beginning with the ERA5 reanalysis, met data are available ever 3 and even ever 1 hour. For comparability reasons between ERA5 and ERAI in this paper, we subsampled ERA5 at the same time resolution as ERAI. In all older reanalysis products (e.g. JRA55), only 6h frequency is available. The are now few places in the manuscript, where critical remarks to our parameterization are included. See P4-5, P17

- *P5: The tracer results are interesting, nice job on the mix of real and synthetic tracers. For example, based on the e90 work, the new mixing in CLAMS-2 is essential in maintaining a clean tropopause. Figure 3, is a very nice representation of the consequences of the mixing.*

  Thanks a lot for this compliment :-)

- *P18 Conclusions. These are reasonable and rational and accurately describe the model results shown here. The AoA spectrum at 350K is interesting and shows a burst of young air with the new parameterizations. What is unclear is whether this carries on to 380-400K. The Hoffman 2022 results sound very interesting, is it limited to the lower stratosphere with very stable layering? Unfortunately, the paper is still being written (and should probably not be used as a reference here).*

  Thanks a lot for this good comment. The double peak structure of the AoA spectrum extends up to 400K. We describe now this point in the revised version. See P16. The reference to the Hoffmann et al work was removed. This paper is almost submitted now.

- *Overall, what I am worried about is that the mixing is set by the modelers based on a type of mixing, but it does not respond to atmospheric physics (did I miss something here). Can the authors get the statistics (e.g., like Tiedtke convective fluxes or BL heights and mixing) from the ERA fields?*

  At the end of the section "Discussion", there are now few remarks to the relation between our mixing approach and physics. See P17-18

[revised manuscript text omitted]

---

## Author Comment (AC2)

Response to reviewer 2

We would like to thank reviewer 2 for a very thoughtful and detailed review of our manuscript that helped to improve the paper. Both reviewers criticized too "technical" notation. Because of this, we improved our notation, simplified Table 2 and tried to avoid techical abbrevations in the main text. Furthermore, a too long chapter 3 was shortened and a new chapter "Discussion" was included. In the last 3 paragraphs of this new chapter, we discuss many of the points raised by both reviewers. In the following, we address all other points (denoted by italic letters). We reference to the changes in the manuscript (marked with red color and attached to this reply) by giving the page number (PXX).

Major comments:

1. *This manuscript is an interesting contribution to the technology of atmospheric transport model, extending the series of works made with CLaMS. The main conclusion is that parameterized convective fluxes are necessary to perform realistic transport from the boundary layer to the UTLS, especially in the tropical region where no conveyor belt is operating, and I ready to accept that. The paper is on the overall well written and I have only a few minor comments to forward to the authors.*

   Thanks a lot for this compliment

Minor comments:

- *My only significant reservation is about the usage of the e90 tracer. Prather et al. (2011) say that the 90 ppbv value fits the tropopause for their given CTM and meteorology but there is no reason to use it as a general reference since there is by definition no observational constrain for such an artificial tracer. It is OK to use e90 in order to characterize the overall structure and time-scale of the tropospheric overturning circulation but I do not think that the comparison of the 90 ppbv surface with the WMO tropopause in Fig.3 can be used to score the versions of the model. In all panels of Fig.2, there are other surfaces with large gradients that fit as well the tropopause than the I1C1 surfaces in Fig.3.*

   Yes, we agree. We use the e90 tracer mainly to visualize the overall differences. To quantify the differences, we apply only in-situ observations as described in section 3.5.

- *The paper is written in an incremental mode and could be somewhat difficult to follow for someone who is not already familiar with CLaMS and the terminology that is proper to this model. Perhaps this is the intended audience but I would find useful to have a few general definition of notions like tropospheric mixing, pure advective mode and so on. The new epsilon parameter is defined is the appendix but this is again very cryptic if you do not know anything about CLaMS.*

   Yes, we completely agree. We simplified the notation (e.g. Table 2) and add few additional explanation. See P4-5. We also simplified the legend in Fig. A3

- *It takes some time to understand and follow the awkward naming convention of the runs. At first, the table 2 does not make sense. Perhaps a more detailed description would help. It is not necessary to name the experiments in a publication in the same way as the directories on the computer.*

Yes, we agree. Table 2 was completely rewritten. We also extended the caption of this Table and add few additional explanation in section 2.

- *The parameterized convection is based on a fairly crude scheme, of the type that has been rejected long ago in the numerical weather forecast to the benefit of more sophisticated schemes which are found necessary to better represent the convective timing and organisation, and the convective fluxes. I understand that the present scheme used in CLaMS is better than nothing but why not using the convective fluxes of the ERA5 which are archived and available along the meteorological data used in this study. This will not be perfect as there are still many biases in such scheme but is would be more in pace with the state of the art in convective parametrization with the sole cost of additional storage. Off course, this suggestion is aimed only at future work.*

  Yes, we completely agree. We discuss all these points in the 3 new paragraphs at the end of the chapter "Discussion". One advantage of the present scheme is, however, that it is independent from the availability of convective fluxes in the driving met. data, and therefore can be easily applied also to other datasets. However, we will certainly improve this point in the future.

- *I would have liked to see a more detailed discussion of the effect of convective parameterization on p. 7, for instance about the meridional gradients associated with the Hadley circulation.*

  Thanks. We now add new sentence related to this point. See P7.

- *I am puzzled by the discrepancies of CO2 in the boundary layer where it is forced by Carbon Tracker. Is it that Carbon Tracker misses the biological carbon cycle and in which way is it local ?*

  Yes, this is also puzzling for us. One reason is that within the assimilation algorithm of CarbonTracker there are not enough stations around Nepal and India as well as around Shannon in Ireland measuring $CO_2$.

- *I do not understand why the black reference dots are different in the various panels of Fig. 6 and I do not understand either why the modelled cloud is closer to the reference in panel b that in panel a with apparently the same values of d. Perhaps this is an effect of the linear color scale that does not display the difference among small values.*

  We think that this point was misunderstood by the reviewer. The reference dots denoting observed $CO$-$O_3$ correlations are color-coded and the CLaMS simulation is plotted with black dots. The colors measure the distance between the reference and the simulated $CO$-$O_3$ correlations. We improved our text to avoid potential ambiguities.

- *There are two modes in Fig. A0 that should be distinguished, a UT tropical mode on the left which is strongly affected by the convection and a LS extra-tropical mode on the right which is only affected by mixing.*

  Great comment. We clarified the discussion and adopted the suggested terminology. See P 16.

- *The authors should check the DOI numbers in the reference list. At least one (Konopka et al., 2019), the one I tried, points to another paper.*

  DOI numbers are checked now.

[revised manuscript text omitted]
 : $\quad \lambda = 4.0 \, \mathrm{d}^{-1}, \quad$ mixing frequency = 6 hours $\quad \epsilon = 0.3$

C1 : $\quad N_m^2 = 0.0 \times 10^{-4} \, \mathrm{s}^{-2}, \quad \Delta\theta_c = 10 \, \mathrm{K}$

[revised manuscript text omitted]